# Solitary Extramedullary Plasmacytoma Presenting as Asymptomatic Palatal Erythroplakia: Report of a Case

**DOI:** 10.3390/ijerph18073762

**Published:** 2021-04-04

**Authors:** Francesca Lalla, Alessandro Vinciguerra, Alessandra Lissoni, Gianluigi Arrigoni, Francesca Lira Luce, Silvio Abati

**Affiliations:** 1Otorhinolaryngology—Head & Neck Surgery Department, San Raffaele Hospital, University Vita-Salute, 20132 Milan, Italy; lalla.francesca@hsr.it (F.L.); liraluce.francesca@hsr.it (F.L.L.); 2School of Medicine, Vita-Salute San Raffaele University, 20132 Milan, Italy; abati.silvio@hsr.it; 3Department of Dentistry and Stomatology IRCCS San Raffaele Hospital, University Vita-Salute, 20132 Milan, Italy; lissoni.alessandra@hsr.it; 4Department of Anatomic Pathology, San Raffaele Hospital, University Vita-Salute, 20132 Milan, Italy; arrigoni.gianluigi@hsr.it

**Keywords:** multiple myeloma, extramedullary plasmacytoma, oral pathology, oral health

## Abstract

Solitary plasmacytoma (SP) is a rare malignant tumor of plasma cells with no systemic spread; however, when it disseminates and affects multiple skeletal sites, it is called multiple myeloma (MM). The etiology of solitary plasmacytoma is unknown, with two possible subtypes: solitary extramedullary plasmacytoma (EMP) and solitary bone plasmacytoma (SBP). We present a case of EMP arising as asymptomatic erythroplakia of the palate, which is rarely described in the literature. The definitive diagnosis was obtained with immunohistochemical studies, after which the lesion was subjected to excisional biopsy. At present, after two years of close follow-up, the patient has shown no signs of relapse or conversion to MM. The uniqueness of the case highlights the possibility of an atypical EMP lesion in the head and neck, thus posing a diagnostic and therapeutic challenge for physicians.

## 1. Introduction

Solitary plasmacytoma is a rare B lymphocytic plasma cell neoplasm without systemic spread and of unknown origin. It can be divided in two groups depending on its location: solitary bone plasmacytoma (SBP) and solitary extramedullary plasmacytoma (EMP). Multiple myeloma (MM) is the systemic form of the pathology, characterized by multifocal disseminated lesions, and represents the most common tumor of plasma cells [1,2]. Both subtypes of solitary plasmacytoma can progress to multiple myeloma, but this occurrence is more common in SPB [3]. SBP accounts for approximately 70% of all cases of solitary plasmacytoma and predominantly affects bones containing red marrow, such as the vertebrae, pelvis, ribs, and femur [4]. It appears as a single, painful, lytic bone lesion due to monoclonal infiltration of plasma cells [4,5]. EMP accounts for just 3% of all plasma cell neoplasms, with a predilection for male subjects between the fourth and sixth decades of life [6,7]. EMP can involve any site or organ, but it is estimated that nearly 80% of cases arise in the subepithelial tissues of the head and neck region, and especially the upper respiratory tract (nose, paranasal sinuses, nasopharynx, and tonsils) for which patients usually experience swelling and moderate pain in the affected area [5,8]. A review of the medical literature revealed that only a small number of patients are reported with EMP of the oral cavity, especially with the involvement of the palate [2,3,6,8,9,10,11]. Some authors have reported the case of a patient presenting with a painless, rapidly growing lesion in the posterior upper alveolar ridge that extended towards the palate, [2] while others described a case of EMP presenting as a massive exophytic lesion of the hard palate that was covered with intact mucosa [8]. Definitive diagnosis is based on the biopsy of the specimen and histopathological examination, while radiotherapy is considered to be the treatment of choice [8]. The aim of this work was to describe an unusual manifestation of EMP and to underline the importance of accurate physical examination and the role of oral biopsy in the diagnostic pathway of oral lesions.

## 2. Case Report

A 45-year-old male with a healthy medical history, who was non-smoker, and who had family history negative for neoplasms presented with a small erythematous spot on the left soft palate that gave him occasional stinging pain. The patient reported the appearance of the lesion 10 months earlier, without any causal traumatic event. Upon physical examination, the erythroplakia was found to be 8 × 4 mm in size and located on the posterior third of the palate, left side next to the midline, in the soft palate close to the limit with the hard palate (Figure 1).

After two weeks with no improvement with a local antiseptic agent, an excisional biopsy was performed (Figure 2).

The differential diagnosis included vascular malformation and macula. After routine staining and kappa and lambda chain immunohistochemistry (IHC), the pathologist made a definitive diagnosis: palatal mucosa with plasma cell infiltrate with clonal restriction for lambda light chains and negative for IgG4, which is diagnostic for plasmacytoma (Figure 3 and Figure 4).

The patient was than referred to the onco-hematologist for further investigations and treatment. Blood tests including full blood count, serum protein electrophoresis (SPEP) and immunofixation, quantitative immunoglobulins, serum-free light chains, urine protein electrophoresis (UPEP), beta-2 microglobulin, calcium, creatinine, and LDH were performed. None of these tests revealed abnormalities compatible with the systemic involvement of the disease; in particular, no monoclonal immunoglobulin, abnormal serum light chains, elevated levels of beta-2 microglobulin or Bence Jones protein were found.

A bone marrow examination included bone marrow biopsy and aspiration; the samples collected through aspiration underwent the following assessments: immunophenotyping, cytogenetics, immunohistochemistry, flow cytometry, and fluorescence in situ hybridization (FISH). The bone marrow biopsy showed negative findings for plasmacytoma, with mild medullary hypoplasia and dyshematopoiesis. Tests performed on the aspirate did not reveal any molecular or genetic anomalies. A PET examination documented a diffuse and symmetrical tracer accumulation in the soft palate, without focal accumulation areas attributable to disease localization. Therefore, considering the surgical radicality and the absence of systemic involvement or suspicious findings for residual disease, the radiotherapist decided that neither external beam RT nor brachytherapy was indicated. Surgical excision followed by close follow-ups—which consisted of onco-hematological and dental check-ups every three months, an MRI every six months, and periodic ENT visits—was considered to be sufficient (Figure 5). Blood tests, bone marrow examination, MRI, and PET confirmed a solitary EMP, since no involvement was found in bone marrow, head and neck segments, or entire skeleton.

At the time of writing, after two years of follow up, the patient has shown no signs of new disease, recurrence, or conversion to multiple myeloma. The patient gave consent for the communication of her case.

## 3. Discussion

The etiology of solitary plasmacytoma is still unknown, with possible roles of viral infections, chronic stimulation, overdose irradiation, and gene disorders in the reticuloendothelial system [6]. The median age at diagnosis is between 55 and 60 years, and approximately 75% of patients are men; our case of EMP was unusual since the patient was less than 50 years old [1,5]. 

EMP can develop in any organ, but in 80–90% of cases, it involves the submucosa of the head and neck area; signs and symptoms are moderate-to-intense pain, swelling, and tissue masses obstructing the respiratory tract or oral cavity [2,12]. When the lesion affects the nasopharynx, the main symptoms are epistaxis, nasal obstruction, and nasal discharge [1,5,13]. Cervical lymph nodes can be affected in 30–40% of cases without representing a worsening of prognosis [8]. The oral cavity is the site most rarely affected by EMP, with swollen, erythematous, or ulcerative lesions, as has been described by several authors: the palate is a site of localization that has only rarely been described (Table 1) [2,3,6,8,10].

Symptoms may vary in relation to the location of the tumor, but the most common complaint is edema [2]. In addition, toothache can be reported in the case of the involvement of dental alveolus or the maxillary sinus, with the destruction of the surrounding bone and damage to adjacent teeth [2,14]. It should be noted that EMP of the oral cavity can present clinical features similar to many benign and malignant lesions of the oral cavity such as pyogenic granuloma, papilloma, lymphoma, squamous carcinoma, and osteosarcoma; for this reason, it is essential to recognize this condition to avoid misdiagnosis [8,15]. Unlike what has been described in the literature, our patient presented a flat, non-ulcerative, erythroplastic lesion of the soft palate without lymph node involvement; the lesion was essentially silent, with the exception of causing a stinging pain that occasionally occurred. 

MM, which represents the systemic form of the disease, has a more severe course and a poor prognosis, thus making differential diagnosis—which is primarily based on the absence of CRAB (increased calcium, renal insufficiency, anemia, or multiple bone lesions) criteria—essential [1,2]. Currently, the accepted diagnostic criteria of EMP include tissue biopsy-indicating monoclonal plasma cell histology, a bone marrow biopsy showing <5% plasma cell with no clonal proliferation, the absence of osteolytic bone lesions or other tissue involvement without proof of myeloma, the absence of hypercalcemia or renal failure, and a low serum or urinary level of monoclonal immunoglobulin [7]. However, the definitive diagnosis of EMP is based on histology, and its specific immunoglobulin secretor type can be determined by IHC: the most common immunoglobulin expressed by tumor cells is IgG, with kappa chain restriction [3,16]. In the presented case, after ineffective local antiseptic therapy, excisional biopsy and IHC were carried out with confirmation for a plasma cell infiltrate with clonal restriction for lambda light chains and negative for IgG4, thus being diagnostic for plasmacytoma. The patient was referred to the onco-hematology service to perform further tests that confirmed the diagnosis of solitary EMP.

Nowadays, RT is considered to the primary treatment for solitary plasmacytoma, with a recommended curative dose of 40–50 Gy over a four-week period, although no guidelines on RT dosage are yet available [1,8]. Surgery is an alternative option for resectable and small lesions, followed by adjuvant RT in cases with positive margins, while chemotherapy is available for recurrent or refractory cases with progression to MM [1,8]. However, given the small size of the lesion and the absence of systemic involvement in our patient, the radiotherapist considered surgical excision followed by close follow-ups to be sufficient. As previously mentioned, the possibility of progression into MM always requires long-term follow up with the measurement of monoclonal immunoglobulin [3]. The detection of monoclonal immunoglobulins in the patient’s serum or urine indicates new disease, recurrence, or conversion to MM [3]. Furthermore, it must be considered that both forms of solitary plasmacytoma have a risk of progression to MM, with a higher risk of progression for solitary bone plasmacytoma; in particular, the risk of dissemination is higher in patients with bulky lesions and IgG positivity with kappa light chain restriction [8,17]. In cases of EMP, the risk of progression is about 10–30% and is independent of lymph node involvement; this occurrence, although rare, requires prolonged follow-ups after adequate therapy [5,8]. In light of the above-mentioned information, we performed close follow-ups with regular dental, onco-hematological, and ENT checks along with periodic MRI, which made it possible to adequately monitor the state of the disease for two years.

## 4. Conclusions

Our experience underlined the diagnostic role of oral biopsy in identifying an atypical presentation of a rare disease. The finding of this rare form of EMP that presented as an asymptomatic palatal erythroplakia highlighted the importance of thorough and complete objective examinations by an oral medicine expert and otolaryngologist, as well as the need to include plasma cell neoplasms in the differential diagnostic pathway of the lesions of the oral cavity in patients of all age groups.

## Figures and Tables

**Figure 1 ijerph-18-03762-f001:**
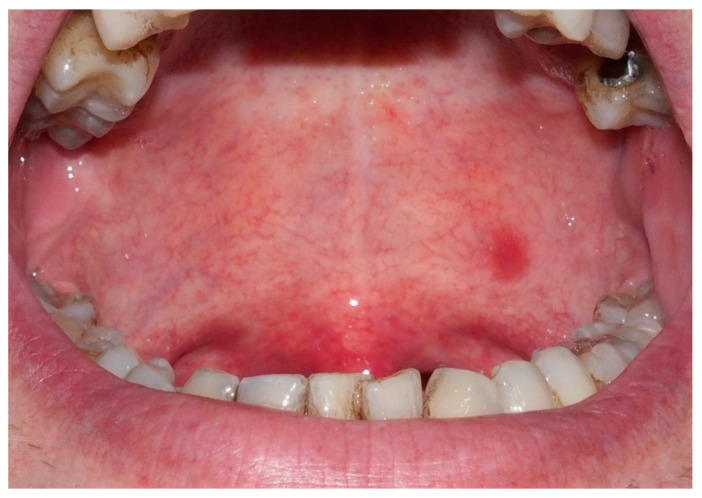
Erythroplastic lesion 8 × 4 mm in size located on the posterior third of the soft palate, left side next to the midline.

**Figure 2 ijerph-18-03762-f002:**
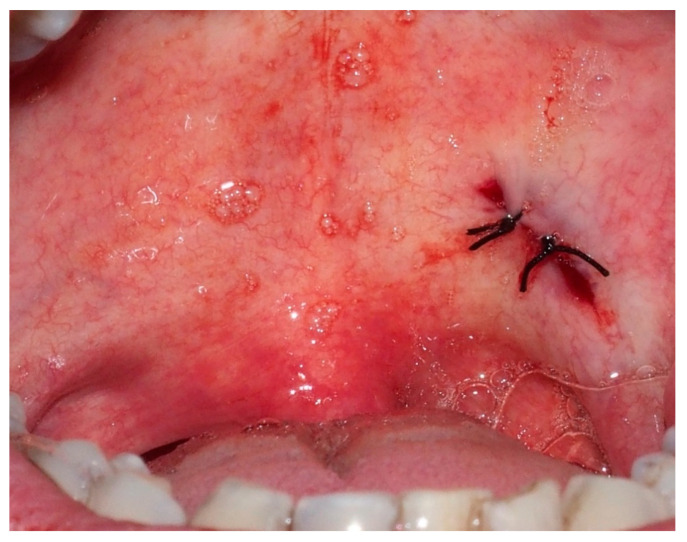
Soft palate after excision of the lesion.

**Figure 3 ijerph-18-03762-f003:**
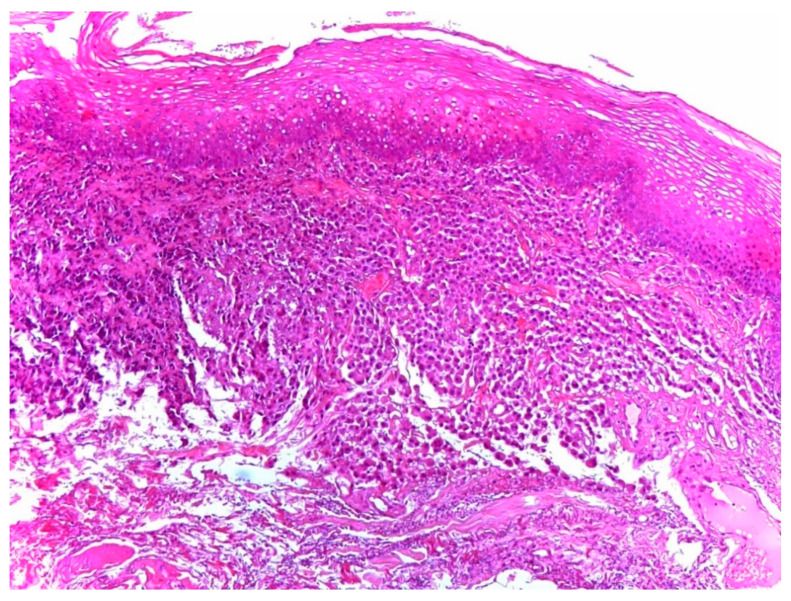
Monomorphic mature plasma cell chorion infiltration (HE 100×/200×)**.**

**Figure 4 ijerph-18-03762-f004:**
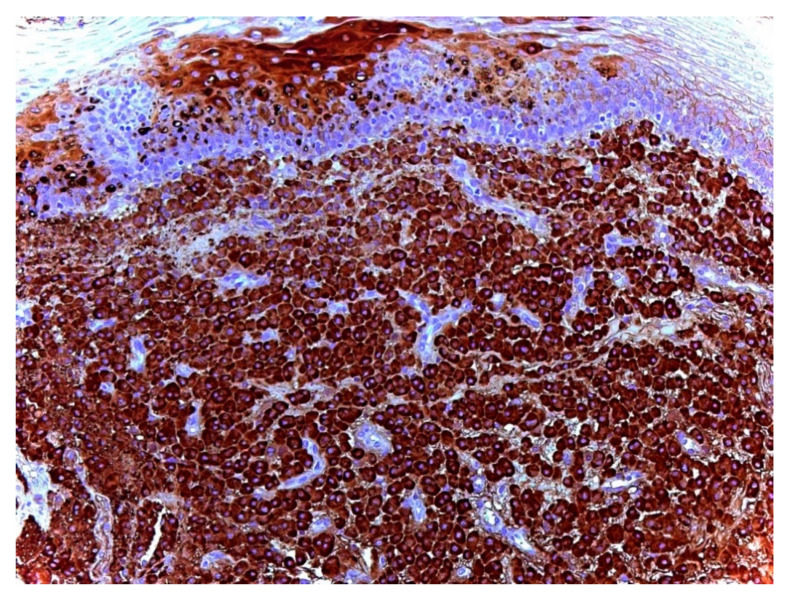
Lambda immunohistochemical expression (monoclonal restriction) in plasma cells (100×/200×).

**Figure 5 ijerph-18-03762-f005:**
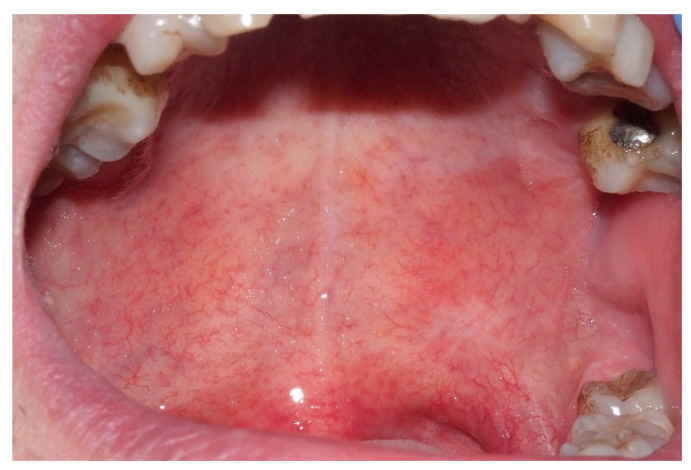
Appearance of the soft palate at one year after the removal of the lesion.

**Table 1 ijerph-18-03762-t001:** Summary of previous reports of extramedullary plasmacytoma (EMP) with the involvement of the palate.

Author	Year	Country	Sex	Age	Location	No of Cases
Webb et al. [9]	1962	USA	F	37	Soft palate	1
Yoshimura et al. [10]	1976	Japan	M	64	Mucosa of the hard palate	1
Susnerwala et al. [11]	1997	UK	M	72	Soft palate	1
Majumdar et al. [3]	2002	UK	M	54	Soft palate, close to the uvula	1
Barros et al. [2]	2011	Brazil	M	70	Lesion in the posterior upper alveolar ridge, extending towards the palate	1
Purkayastha et al. [8]	2016	India	M	44	Hard palate	1
Gholizadeh et al. [6]	2016	Iran	M	25	Palatal side of left maxillary second and third molars	1

## Data Availability

The data presented in this study are available on request from the corresponding author. The data are not publicly available due to privacy restrictions.

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
