# Peer review of "Solitary Extramedullary Plasmacytoma Presenting as Asymptomatic Palatal Erythroplakia: Report of a Case"

_ijerph, 2021, doi:10.3390/ijerph18073762_

Round 1

Reviewer 1 Report

We present a very interesting case of solitary plasmacytoma with 2 years of follow-up and a good histopathological study.

It would be advisable to introduce in the title ": report of a case."

The introduction should be more extensive with an exhaustive review of the published cases of this type so far.

The description of the case is very well presented. It should be added that "the patient gave consent for the communication of her case."

At the end of the introduction you should put the purpose of this publication.

Author Response

Response to Reviewer 1 Comments

Dear reviewer,

Thank you for the time and effort you have dedicated to providing insightful feedback on ways to strengthen our paper. Your comments were very helpful overall, and we are appreciative of such constructive feedback on our original submission. We have incorporated changes that reflect the detailed suggestions you have graciously provided and we hope that our edits satisfactorily address all the issues and concerns you have noted.

In particular, we inserted the expression "report of a case" in the title and we added that the patient gave consent for the communication of her case.

The introduction has been enriched by relevant references of similar clinical cases published to date; a table with the cases of extramedullary plasmacytoma of the palate described in the literature to date was also included in the discussion. Furthermore, the purpose of the work was inserted at the end of the introduction.

Reviewer 2 Report

Extramedullary plasmacytoma is very rare, however important presentation of solitary plasmacytoma. It is because it has a 30% risk of developing symptomatic Multiple myeloma within 10 years after the initial diagnosis.

Case report is well prepared, however some detailed information  has to be provided, important for diagnosis and  prognosis:

  1. What blood tests was done?
  2. What were the outcomes?
  3. What included “bone marrow examination”?

PET/CT is recommended for patients with EMP to detect additional soft tissue lesions, it has also therapeutic implications.

  1. Why despite positive PET after surgical excision standard radiotherapy was not implemented?
  2. Why despite prognostic value PET was not performed during observation period?

Important for the reader will be information about follow-up.

  1. What does “close follow-up mean”?
  2. Whether other recommended tests and examinations were performed besides mentioned ones?
  3. What ENT visit includes?

Author Response

Response to Reviewer 2 Comments

Dear reviewer, thank you for your comments. Below you can find the point-by-point response

Point 1: What blood tests was done?

Response 1: Thank you for your question. The blood tests performed were full blood count, serum protein electrophoresis (SPEP) and immunofixation, quantitative immunoglobulins (IgG, IgA, IgM), serum free light chains (with light chain ratio), urine protein electrophoresis (UPEP), beta- 2 microglobulin, calcium, creatinine, LDH.

As you kindly requested, we have added the above information in the text.

(revised submission, page 4, lines 80-83)

Point 2: What were the outcomes?

Response 2: Blood and urine tests performed in the preoperative phase and in the follow up did not reveal any abnormalities compatible with systemic involvement of the disease, in particular no monoclonal immunoglobulin, abnormal serum light chains, elevated levels of beta-2 microglobulin or Bence Jones protein were found. We have added such information following your precious question.

(revised submission, page 4, lines 83-86)

Point 3: What included “bone marrow examination”?

Response 3: You make a fair assessment. The bone marrow examination included bone marrow biopsy and aspiration; the samples collected through aspiration underwent the following assessments: immunophenotyping, cytogenetics, immunohistochemistry, flow cytometry and fluorescence in situ hybridization (FISH). The bone marrow biopsy showed negative findings for plasmacytoma, with mild medullary hypoplasia and dyshematopoiesis. Tests performed on the aspirate did not reveal any molecular or genetic anomalies. We have added the above information in the text, following your valuable indications.

(revised submission, page 4, lines 87-92)

Point 4: Why despite positive PET after surgical excision standard radiotherapy was not implemented?

Response 4: Thank you for noting this. Radiotherapy (RT) is considered the primary treatment for solitary plasmacytoma, while adjuvant RT following surgical excision is indicated for small lesions with positive margins. In our case, the PET examination documented a diffuse and symmetrical tracer accumulation in the soft palate, without focal accumulation areas attributable to disease localization. Therefore, considering the surgical radicality and the absence of systemic involvement or suspicious findings for residual disease, the radiotherapist considered neither external beam RT nor brachytherapy indicated; surgical excision followed by close follow-up was considered to be sufficient. We have added such information in the text, following your precious question.

(revised submission, page 4, lines 96-101)

Point 5: Why despite prognostic value PET was not performed during observation period?

Response 5: Thank you. As well known, PET is indicated in patients with EMP to detect further soft tissue lesions. In this case, after surgical excision of the lesion, a PET scan was performed which did not reveal further lesions of the adjacent or distant soft tissues. In consideration of the absence of systemic involvement, the small size of the lesion and the surgical radicality of the treatment, MRI was considered an adequate imaging follow-up for local and regional recurrence, providing precise anatomical and functional data. Nevertheless, PET has a prognostic value and provides information about the response to treatments: hence, the follow up, still in progress, will also include periodic PET scans.

Point 6: What does “close follow-up” mean?

Response 6: Thank you for your question. The patient performed a close follow up, represented by blood tests and clinical evaluation (haematological and dental check-ups) every 3 months, ENT visit every 3-4 months and MRI every 6 months. The follow-up is still ongoing and will include periodic PET tests, in addition to the exams already mentioned. We have included this information in the text, in response to your valuable observation

(revised submission, page 4, lines 101-103)

Point 7: Whether other recommended tests and examinations were performed besides mentioned ones?

Response 7: In consideration of the absence of clinical and laboratory signs of systemic spread of the disease and of the substantial surgical radicality of the treatment, no further tests other than those mentioned were performed.

Point 8: What ENT visit includes?

Response 8: Thank you for your question. The ENT visit included a thorough inspection of the nose, ears, oral cavity and the search for any suspicious laterocervical lymph nodes. The study of the larynx was performed with a flexible fiberoptic laryngoscope.

Round 2

Reviewer 1 Report

The manuscript has improved with the changes made and makes the objective clear, as well as the case description section improves. Mention the consent of the patient for the communication of his case. Improvements have also been made to the references section.